# Where's Waldo: Diffusion Features For Personalized Segmentation and Retrieval

**Dvir Samuel**[1,2]*    **Rami Ben-Ari**[2]    **Matan Levy**[3]    **Nir Darshan**[2]    **Gal Chechik**[1,4]

[1]Bar-Ilan University, Israel
[2]OriginAI, Israel
[3]The Hebrew University of Jerusalem, Israel
[4]NVIDIA Research, Israel

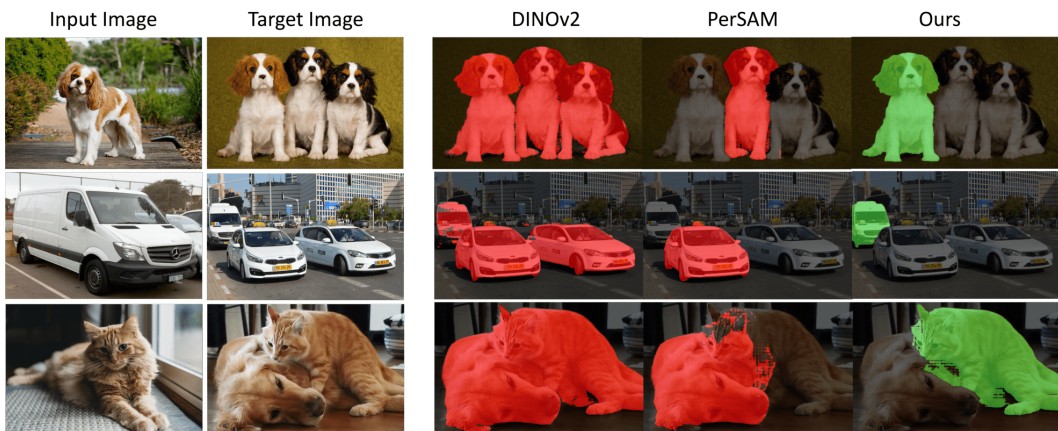

Figure 1: *Personalized* segmentation task involves segmenting a specific reference object in a new scene. Our method is capable to accurately identify the specific reference instance in the target image, even when other objects from the same class are present. While other methods capture visually or semantically similar objects, our method can successfully extract the identical instance, by using a new personalized feature map and fusing semantic and appearance cues. Red and green indicate incorrect and correct segmentations respectively.

## Abstract

Personalized retrieval and segmentation aim to locate specific instances within a dataset based on an input image and a short description of the reference instance. While supervised methods are effective, they require extensive labeled data for training. Recently, self-supervised foundation models have been introduced to these tasks showing comparable results to supervised methods. However, a significant flaw in these models is evident: they struggle to locate a desired instance when other instances within the same class are presented. In this paper, we explore text-to-image diffusion models for these tasks. Specifically, we propose a novel approach called PDM for **P**ersonalized **D**iffusion Features **M**atching, that leverages intermediate features of pre-trained text-to-image models for personalization tasks without any additional training. PDM demonstrates superior performance on popular retrieval and segmentation benchmarks, outperforming even super-

*Correspondence to: Dvir Samuel <dvirsamuel@gmail.com>.

38th Conference on Neural Information Processing Systems (NeurIPS 2024).

vised methods. We also highlight notable shortcomings in current instance and segmentation datasets and propose new benchmarks for these tasks.

# 1   Introduction

Personalized retrieval and segmentation focus on identifying specific instances within a dataset. When provided with an input image featuring a particular instance (such as your beloved cat) and a brief description ("A cat"), the objective is to locate and segment this exact instance throughout a large collection of images. Personalized methods are useful in various applications, including instance search [34], product identification [6, 41], and landmark recognition [47]. Furthermore, personalized segmentation can be applied to video tracking [45], automatic labeling [43], and image editing [5, 8].

While supervised methods can be effective for these tasks, they require an extensive amount of labeled training data. Recently, a self-supervised foundation model was proposed [45] to address this task. This model uses the SAM encoder [14] or DINOv2 [23] foundation model to extract spatial features from a given reference instance. These features are then used to localize the object instance in the target image. While effective when a single instance appear in the target image, both DINOv2 and SAM fall short when multiple instances within the same object class are presented in the image. This is illustrated in Figure 1 showing failure cases of DinoV2 and SAM in localizing the correct dog or van (see first and second row). They also fail when two similar objects from different semantic classes are presented (wrongly segmenting the dog instead of the cat.)

In this paper, we propose to explore text-to-image diffusion models for these tasks. Text-to-image foundation models have achieved remarkable success in generating new and unique images from text prompts [7, 30, 31, 33]. These models have the capability to generate an infinite array of objects and instances, each exhibiting unique appearances and structures. Consequently, it is reasonable to hypothesize that properties of generated objects are encoded within the intermediate features of the diffusion model during generation. Recent studies [1, 37, 40] show zero-shot capabilities to create subtle changes in generated instances by manipulating the intermediate activation of the diffusion layers, during generation. Although effective, using text-to-image diffusion models "out of the box" for instance-related tasks, beyond generation or editing, remains unexplored.

In this paper, we present a new approach, called **PDM**, **P**ersonalized **D**iffusion Features **M**atching, for personalized retrieval and segmentation. PDM requires no training or fine-tuning, no prompt optimization, or any additional models. We demonstrate how a specific layer and block contain hidden textural and semantic information. These features are then used for the localization of a reference instance within a given target image, enabling both personalized segmentation and retrieval. PDM builds upon these newly discovered diffusion features, and surpasses other self-supervised methods (like DINOv2 [23], SAM [14] and DIFT [36]) weakly supervised methods (CLIP, OpenCLIP) and even supervised methods on personalized instance retrieval and segmentation tasks.

We also address significant limitations in traditional benchmarks for retrieval and segmentation. Current benchmarks often feature images with a single, distinct object or multiple objects from different categories, allowing semantic-based methods to achieve high accuracy. To overcome these deficiencies, we construct new benchmarks based on a newly published video tracking and segmentation dataset [4]. This dataset includes videos with multiple instances from the same category (e.g. two dogs playing or a group of people talking). Our method significantly outperforms all baselines on this new dataset, highlighting its ability to accurately handle multiple similar instances and demonstrating its superior capability in personalized retrieval and segmentation.

# 2   Related Work

**Exploring pre-trained diffusion features.** Text-to-image diffusion models [7, 30, 31, 33] have demonstrated state-of-the-art performance for image generation tasks. With its superior generation ability, recent studies started investigating the internal representation of diffusion models. DIFT [36] and Fuse [44] showed that extracting features from the ResNet layers of the denoising module provides a semantic correspondence between two objects which can also be used for image editing propagation. Plug-and-Play [40] suggested to extract features from self-attention layers of a reference image, during the image generation process, while incorporated with a text prompt. This approach

showcased that output images can retain the structure of the reference image while embodying the appearance described in the text prompt. Cross-Image-Attention [1] further showed that sub-layers in self-attention layers correspond to the structure and the appearance of generated images. Their findings enabled the generation of images that blend the structure from one image with the appearance from another. ConsiStory [37] recently suggested injecting the self-attention features of an instance from a pre-generated image into the generation process of other images to ensure consistent reproduction of the same instance across images. DiffSeg [38] introduced a method using self-attention maps for zero-shot image segmentation. They aggregate attention maps from multiple self-attention layers during image generation and merge them iteratively to produce a stack of object proposals. Segmentation maps are then obtained by applying Non-Maximum Suppression over the merged maps. In contrast to these studies, in this paper, we explore using internal features of pre-trained diffusion models for instance related tasks.

**Personalized Segmentation:** PerSAM [45] introduced the use of SAM [45] for personalized image segmentation. They employed the SAM [14] encoder (or DINOv2 [23]) for the representation of the reference and target images, which are then used to calculate a confidence map localizing the user's reference instance in the target image. Finally, it predicts positive and negative points on the target image to be used as prompts for SAM. Additionally, they proposed a new benchmark, called *PerSeg*, for personalized image segmentation. It includes 40 objects across various categories, each associated with 5-7 images, and is evaluated using mIoU and bIoU metrics.

**Instance Retrieval:** Content-based instance retrieval can be seen as a variant of personalized retrieval where images contain only a single instance. Recent supervised methods, GSS [21] and HP [2] proposed Graph Networks for effective retrieval. SuperGlobal [34] proposed a memory-efficient image retrieval method, that specifically focuses on the global feature extraction while in the re-ranking stage, they update the global features of the query and top-ranked images by only considering feature refinement with a small set of images, thus being very efficient. Recently, also self-supervised models [9, 11, 23, 46] show comparable performance to supervised methods on retrieval tasks. These techniques achieve impressive results in zero-shot scenarios however, they often necessitate model fine-tuning to achieve optimal performance. In this study, we investigate text-to-image diffusion models, which belong to the category of self-supervised models, for zero-shot personalized retrieval and segmentation tasks. Our findings show that diffusion features supress features from other self-supervised foundation models.

**Semantic-level Feature Matching.** Recent works have focused on improving semantic-level feature matching in various tasks. SIGMA [19] introduces semantic-complete graph matching for Domain Adaptive Object Detection, addressing within-class variance through node-to-node matching. Light-Glue [20] enhances local feature matching efficiency with a deep network adaptive to image difficulty, making it ideal for latency-sensitive tasks. In this paper, we propose Personalized Diffusion Features Matching (PDM), which leverages intermediate features from pre-trained text-to-image diffusion models for personalized retrieval and segmentation without additional training.

## 3 Method

In this section, we describe our approach to leverage pre-trained diffusion models for personalized retrieval and segmentation. We begin by defining these tasks and then delve into identifying features that encompass both semantic and appearance aspects. Lastly, we demonstrate the application of these features in personalized instance retrieval and segmentation.

### 3.1 Personalized Retrieval and Segmentation.

In personalized retrieval and segmentation, the user supplies a single reference image, and a mask indicating the reference instance [45] or the class name of the instance [10]. This work focuses on the case where only class names are provided. For personalized retrieval, the goal is to retrieve images from a database that contains the exact instance specified in the reference image. In personalized segmentation, the objective is to segment the specified instance in new images and videos.

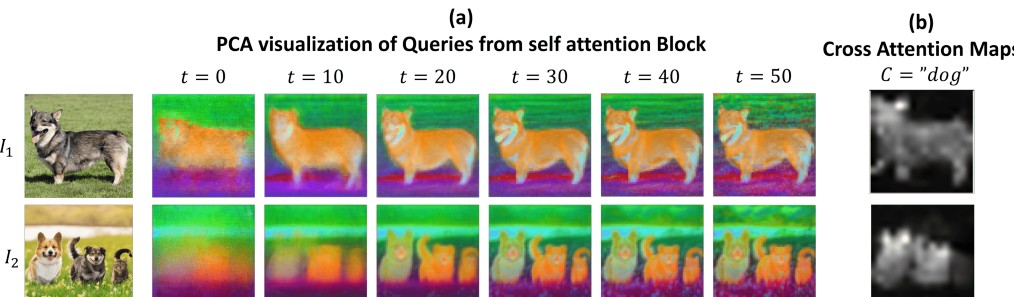

Figure 2: (a) PCA visualization of $\mathcal{Q}^{\mathbf{S}A}$ features obtained from the first self-attention block in the last layer of the U-Net module, at various diffusion timesteps. Objects with similar textures and colors have similar features. The dog's color in $I_1$ is similar to the colors of both the dog and the cat in $I_2$, indicating textural similarity. Additionally, the localization is sharper at larger timesteps. (b) Visualization of the cross-attention map $\mathcal{F}^S \mathcal{C}^T$ for a given prompt "dog". Note the higher region correlation (brighter colors) corresponding to the dog, while overlooking the cat in the bottom image.

## 3.2 Are instance features even encoded in a pre-trained text-to-image model?

Pre-trained text-to-image models can generate an endless variety of objects and instances, each with unique visual characteristics and structures. Recent methods have demonstrated that specific changes in the activations of self and cross-attention activations of the diffusion layer can influence the appearance of specific instances in the generated image. These methods typically modify all activations across all denoising timestamps to affect the generated image. This indicates that instance features are indeed encoded within these models. One can propose to use all diffusion activations during generation and aggregate them for downstream tasks. However, using all features extracted from diffusion layers is memory-intensive and computationally demanding. It also raises the challenge of merging all these features coherently.

We aim to identify *a single layer* at a unique timestamp where both the semantics and appearance (texture) of a reference instance are encoded. We first briefly explain how we extract features from Stable Diffusion [31], a pre-trained text-to-image model. The architecture of Stable Diffusion consists of a VAE encoder and a VAE decoder that facilitates the conversion between the pixel and latent spaces, and a denoising U-Net module that operates in the latent space. We refer the reader to Appendix A, for preliminary on the internal structure of the denoising U-Net layer. We first encode input image $I$ into the latent space of a VAE using an encoder to produce a latent code $z_0$. Next, we employ a diffusion inversion method [24, 35], to compute the latent code $z_t$ at the time step $t$ with the class name embedding as inputs. We then run denoising step at timestamp t to extract activations (features) from the denoising U-Net.

Previous studies [36, 40, 44] observed that outputs of earlier layers from the U-Net decoder capture coarse yet consistent semantic correspondences, while deeper layers capture more low-level details and high-frequency information. Based on these observations, and in contrast to previous work, we conducted a more thorough analysis of features extracted from *all blocks* of the *last* U-Net layer, examining their role across different timestamps. Interestingly, we consistently found that appearance features are encoded in the queries ($\mathcal{Q}^{SA}$) and keys ($\mathcal{K}^{SA}$) matrices of the self-attention (SA) block. This is illustrated in Figure 2(a), where we perform Principal Component Analysis (PCA) on features extracted for a pair of images, at various timestamps. It shows that $\mathcal{Q}^{SA}$ features of the dog in $I_1$ are similar (same color and texture) to those of the middle dog and cat in $I_2$, indicating that textural features are encoded in these layers (similar results are observed for $\mathcal{K}^{SA}$ features).

We therefore define *appearance features* of an image to be the average tensor of $\mathcal{Q}^{SA}$ and $\mathcal{K}^{SA}$ features with dimensions $h \times w \times d$ extracted from the **self**-attention (SA) block, at the last layer $L$ of timestamp $t$:

$$\mathcal{F}^A = \frac{1}{2}(\mathcal{Q}_t^{\mathbf{S}A(L)} + \mathcal{K}_t^{\mathbf{S}A(L)}) \in \mathbb{R}^{h \times w \times d}. \tag{1}$$

Here, $h$ and $w$ represent spatial resolutions of features extracted from layer $L$, while $d$ denotes the feature dimension.

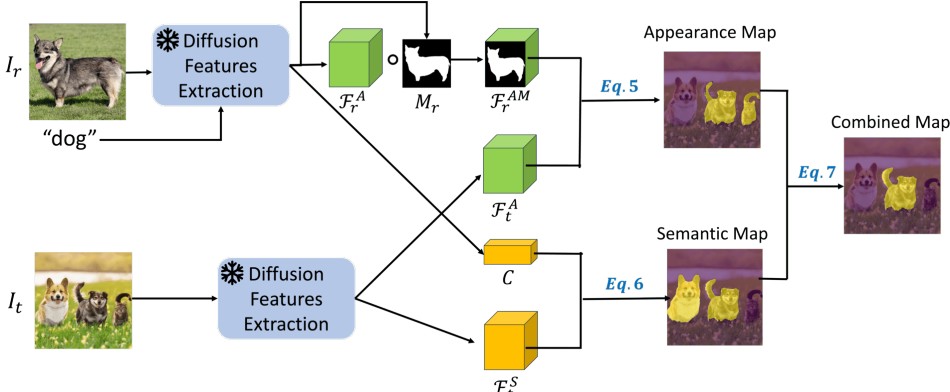

Figure 3: An overview of our **P**ersonalized **D**iffusion Features **M**atching approach. PDM combines semantic and appearance features for zero-shot personalized retrieval and segmentation. We first extract features from the reference, $I_r$ and target $I_t$ images. Appearance similarity is determined by dot product of cropped foreground features from the reference feature map, $\mathcal{F}_r^{AM}$ and the target feature map $\mathcal{F}_t^A$ (Eq. 5) . Semantic similarity is calculated as the product between class name token $\mathcal{C}$ and the target semantic feature map $\mathcal{F}_t^S$ to create a Semantic Map (Eq. 6). The final similarity map $S^{DF}$ combines both maps by average pooling. Note, that while the appearance and semantic maps attend on two dogs, their fusion yields a single and correct result.

For semantic similarity, [16] observed that cross-attention maps establish the relationship between the textual input prompt and patch/pixel-wise image features, effectively allowing a coarse semantic segmentation map that highlights areas of potential object localization. This is further illustrated in Figure 2(b), where the cross attention of the word "dog" with both images results in an attention map highlighting the location where dogs can be found. This cross-attention map is calculated by fusion of the spatial feature map and the token embedding, after projection. Therefore, we define *the semantic features* to be the projected spatial features of the **cross**-attention (CA) block:

$$\mathcal{F}^S = \mathcal{Q}_t^{\mathbf{C}A(L)} \in \mathbb{R}^{h \times w \times d}. \tag{2}$$

### 3.3   Personalized Diffusion Features Matching

We now describe our method for combining semantic and appearance features to address personalized retrieval and segmentation tasks in a zero-shot manner, without training or fine-tuning. We call our approach **PDM** for **P**ersonalized **D**iffusion Features **M**atching. See Figure 3 for illustration.

Let $\mathcal{F}_r^A$, $\mathcal{F}_r^S$ and $\mathcal{F}_t^A$, $\mathcal{F}_t^S$ denote the *appearance* and *semantic* features extracted for the *reference* image $I_r$ and *target* image $I_t$ respectively. Next we define our appearance and semantic similarity functions.

**Appearance Similarity:** We start by localizing objects in the target image that have similar visual features as the reference instance in $I_r$. To this end, we make use of $\mathcal{C} \in \mathbb{R}^{1 \times d}$ as the projected token vector of the class name, extracted from the cross-attention block CA(L)(same block as $\mathcal{F}^S$).

We first use the cross-attention map between spatial image features and $\mathcal{C}$ to obtain a reference mask $M_r$. Specifically:

$$M_r = \mathbb{I}(softmax(\frac{\mathcal{F}_r^S \mathcal{C}^T}{\sqrt{d}}) > \tau) \in \mathbb{R}^{h \times w}. \tag{3}$$

This mask is used to crop relevant appearance features of the instance from the feature map $\mathcal{F}_r^A$, which will later be used for searching within target images. The masked appearance feature map is thus defined as:

$$\mathcal{F}_r^{AM} = M_r \circ \mathcal{F}_r^A \tag{4}$$

$\mathbb{I}$ is the indicator function and $\tau$ is a threshold, resulting eventually in a binary mask, with $n$ foreground features (discarding zeroed-out tokens). Note that $\circ$ denotes spatial-wise multiplication. This approach leverages the U-Net's ability to preserve spatial information in its latent codes and

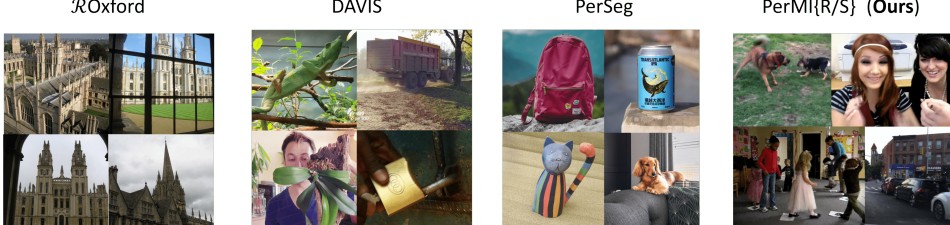

| $\mathcal{R}$Oxford | DAVIS | PerSeg | PerMI{R/S} (**Ours**) |

Figure 4: Examples of personalized retrieval and segmentation benchmarks. Current benchmarks mostly show one single instance in an image or multiple instances from different object classes. Our benchmark for both retrieval and segmentation introduces a realistic and challenging case where multiple instances from the same object class are in the image, *e.g.* two dogs or multiple cars.

features during the diffusion process. Next, we compute a map for the *appearance* similarity score between the reference and target image by simply applying a dot product between the corresponding masked reference feature map and target feature map, followed by average pooling:

$$\mathcal{S}^A = \frac{1}{n} \sum_{i=1}^{n} \mathcal{F}_r^{AM}(i) \cdot \mathcal{F}_t^A \tag{5}$$

where $\mathcal{S}^A \in \mathbb{R}^{h \times w}$ and $\mathcal{F}_r^{AM}(i)$ refers to the feature map $i$ in $\mathcal{F}_r^{AM}(i)$.

**Semantic Similarity:** Here we would like to localize all objects that have the same semantic category as the reference instance. To achieve this, we make use of the semantics encoded in the input class name and calculate a score map between $\mathcal{C}$ and $\mathcal{F}_t^S$. Specifically, we compute:

$$\mathcal{S}^S = \mathcal{F}_t^S \, \mathcal{C}^T \tag{6}$$

The overall diffusion feature (DF) score map combining both semantic (conceptual) and appearance (textural) features is then

$$\mathcal{S}^{DF} = \frac{1}{2}(\mathcal{S}^A + \mathcal{S}^S) \in \mathbb{R}^{h \times w}. \tag{7}$$

**Using diffusion features for personalized retrieval and segmentation.** For personalized retrieval, we rank the target images, using a global score, obtained from the average of $\mathcal{S}^{DF}$, indicating the matching score between a target (candidate) and the reference (query) image. For personalized segmentation, we propose two variations: (1) The score map $\mathcal{S}^{DF}$ is upsampled to the size of the target image, using a binary threshold. We then segment all pixels that are above that threshold. (2) Following [45], we select the point with the highest confidence value in $\mathcal{S}^{DF}$ as *positive* prompt for the position of the target object, and use it to segment the object with SAM [14].

## 4 Evaluation Datasets for Personalized Retrieval and Segmentation

For the evaluation of PDM, we adopted traditional instance retrieval and one-shot segmentation benchmarks, where we also used the provided class names. While preparing these benchmarks, we discovered that most existing instance retrieval and one-shot segmentation benchmarks predominantly showcase only a single instance per object class. For instance, widely used instance retrieval benchmarks such as $\mathcal{R}$Paris [28] and $\mathcal{R}$Oxford [28], focus on single landmarks in their images, with categories typically representing only one possible instance. Similarly, image and video segmentation benchmarks such as the popular Davis [27] dataset and PerSeg [45] mainly comprise either a single instance or multiple instances from diverse object classes, each exhibiting distinct visual and semantic characteristics. This is illustrated in Figure 4. These trends make it relatively straightforward for semantic-based methods to accurately retrieve or segment instances, as there are often no hard negative instances (objects from the same category but different instance) within or across images. Consequently, comparing instance-based features with current methods on such benchmarks often yields comparable results, failing to highlight the strengths of instance-based methods.

To establish a clear distinction between semantic-based and instance-level methods, we introduce two new benchmarks: **Personalized Multi-Instance Retrieval** (**PerMIR**) and **Personalized Multi-Instance Segmentation** (**PerMIS**). Our proposed benchmarks are constructed using the recently

introduced BURST dataset [4], which serves for Object Recognition, Segmentation, and Tracking in Video. This dataset contains videos with pixel-precise segmentation masks for all unique object tracks spanning different object classes. As the dataset encompasses both single-instance and multi-instance videos, we focus on videos containing at least one hard negative instance per video. Specifically, we select videos with a minimum of two instances belonging to the same object class. We then filter out frames that do not contain these instances. This filtering process results in 150 videos across 16 object classes, with an average of 3.1 instances per frame. Detailed statistics can be found in Appendix C. Finally, for the personalized instance retrieval (PerMIR), we randomly chose three frames from each video, designating one as the query frame and the remaining two as the database (gallery) frames. For the personalized image segmentation task (PerMIS-image) we randomly pick three frames from every video, assigning one as the query frame and the others for evaluation. Ground-truth masks are used for cropping the instance from query images and are also used for segmentation evaluation. We further evaluate on the task of video label propagation. For this task we use the first frame of a video as the reference image and the subsequent frames for evaluation. We intend to make our generated datasets publicly available for future work.

# 5 Experiments

We evaluate PDM across three main tasks: (1) **Personalized image and video segmentation**, (2) **Personalized retrieval** and (3) **Video label propagation** where a single video frame is given with object segmentation and the aim is to propagate labels (masks) across video frames, leveraging the information provided by previous frames. The ablation study can be found in Appendix B

**Implementation details.** The main bottleneck of PDM is the real image inversion process, where the image is converted to its noise latent representation for subsequent feature extraction. Using SoTA inversion technique by [24] with Vanilla StableDiffusion, takes about 5 seconds for each image on a single A100. This is due to the requirement of 50 inversion steps. In order to mitigate this, we integrated [24] into SDXL-turbo, a variant of stable diffusion requiring only 4 inversion steps. This decreases the inversion time to $0.5$ seconds per image. Therefore, for all our experiments, features were extracted from SDXL-turbo at the last U-Net layer at the first timestep $t = 4$. Furthermore, all images were resized to 512 x 512 for proper image inversion. We set $\tau$, the threshold for $M_r$ to be $0.7$ for all our experiments.

## 5.1 Personalized Image Segmentation

**Datasets.** We conducted experiments across two personalized (one-shot) image segmentation benchmarks. We first evaluate PDM on the PerSeg [45] dataset, which comprises 40 objects spanning diverse categories such as daily necessities, animals, and buildings. Each object is represented by 5-7 images and masks, capturing different poses or scenes. Additionally, we assessed our method's performance on the PerMIS-Image benchmark (Section 4).

**Baselines.** We evaluate our method by contrasting it with different self-supervised foundation models: (1) DINOv2 [23], (2) PerSAM [45], (3) DIFT [36] and DiffSeg [38]. Additionally, we benchmark it against SoTA-supervised techniques trained specifically for image segmentation, namely SEEM [48] and SegGPT [42].

**Evaluation protocol.** Following [23, 36], we report mIOU and bIOU metrics over all benchmarks. Segmentation with PDM is done by upsampling $\mathcal{S}^{DF}$ to image size. Segmentation with DINOv2 and DIFT is done using features as a similarity function. Specifically, nearest neighbors are found between the query features and target gallery features. No training is involved. We additionally report results with SAM integration, as proposed by [45] (see 3). Here, features are utilized to derive a positive point, followed by segmentation using SAM.

**Results.** Table 1a presents the results of our experiments in personalized image segmentation. Our approach, denoted as **ours**, outperforms supervised methods trained specifically for image segmentation. Additionally, our method achieves superior performance compared to other self-supervised models, including DINOv2 [23], DIFT [36], and PerSAM [45]. We also demonstrate a significant improvement in performance by applying PerSAM with our method, called PerSAM(PDM), surpassing both benchmarks by a considerable margin. Figure 5(a) provides qualitative segmentation results showing that our method reliably identifies the reference instance despite substantial variations in

Table 1: Benchmark **(a) Personalized Segmentation (b) Video Label Propagation**. Our method shows the best performance on all benchmarks and achieves a notable balance between $J$ and $F$, indicating its effectiveness in capturing both region and contour details.

| | (a) Perosnalized Image Segmentation | | | | (b) Video Label Propogation | | | | | |
|---|---|---|---|---|---|---|---|---|---|---|
| | PerSeg | | PerMIS (Image) | | DAVIS | | | PerMIS (Video) | | |
| Model | mIoU | bIoU | mIoU | bIoU | $J\&F$ | $J$ | $F$ | $J\&F$ | $J$ | $F$ |
| SEEM [48] | 87.1 | 55.7 | 14.3 | 35.8 | - | - | - | - | - | - |
| SegGPT [42] | 94.3 | 76.5 | 18.7 | 39.5 | - | - | - | - | - | - |
| MAST [15] | - | - | - | - | 65.5 | 63.3 | 67.6 | 65.1 | 61.7 | 69.2 |
| SFC [12] | - | - | - | - | 71.2 | 68.3 | 74.0 | 73.2 | 70.2 | 76.3 |
| DINOv2 [23] | 68.7 | 27.6 | 20.2 | 41.9 | 71.4 | 67.9 | 74.9 | 5.4 | 62.5 | 68.6 |
| DIFT [36] | 63.2 | 26.9 | 21.9 | 43.1 | 70.0 | 67.4 | 78.6 | 69.7 | 67.3 | 71.8 |
| DiffSeg [38] | 38.6 | 37.9 | 7.9 | 6.4 | - | - | - | - | - | - |
| PerSAM(SAM) [45] | 95.3 | 77.9 | 16.5 | 38.3 | 76.1 | 74.9 | 79.7 | 64.0 | 61.8 | 67.1 |
| PDM (**ours**) | **95.4** | **79.8** | **42.3** | **86.8** | 75.8 | 72.9 | **80.1** | 75.1 | 72.1 | 78.0 |
| PerSAM(PDM) (**ours**) | **97.4** | **81.9** | **49.7** | **89.3** | **78.0** | **75.1** | **81.9** | **76.5** | **73.5** | **79.4** |

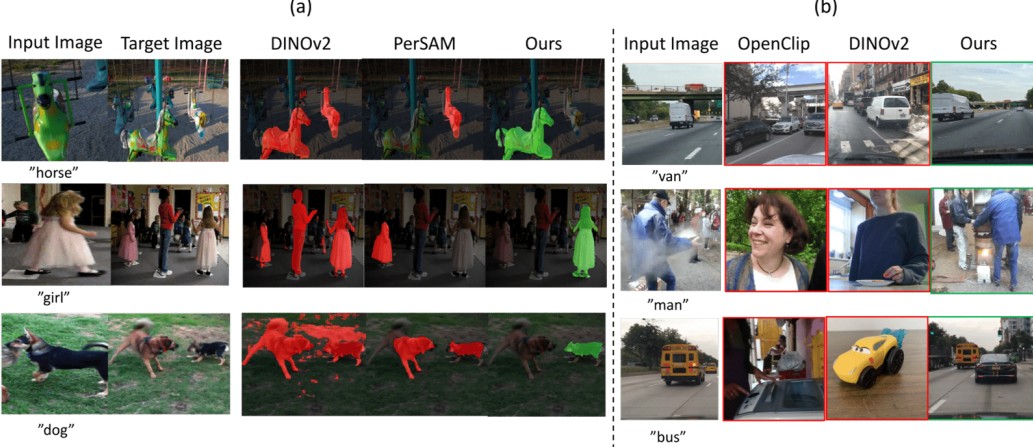

Figure 5: **Qualitative Comparison:** (a) Personalized Segmentation: Red and green indicate incorrect and correct segmentation, respectively. Our method accurately recognizes the reference instance despite significant variations (view angle, pose, or scale), while other methods often capture false positives from the same category. (b) Image Retrieval: Top-1 retrieved image is shown for each method. Note how our model identifies images containing the same instance, despite their small size and large variations. Other methods tend to capture only semantic similarity. Retrieval images have been zoomed in and cropped for clarity.

the target image, whereas other methods frequently capture false positives within the same category. Additional qualitative results in Appendix F

## 5.2 Video Label Propagation

**Datasets.** We further conducted experiments across two temporal one-shot image segmentation benchmarks. We conducted evaluations on the DAVIS17 dataset [27]. This dataset comprises 150 video sequences, with object masks provided for all frames. Furthermore, we evaluated our method's performance on the PerMIS-Video benchmark (Section 4).

**Evaluation protocol.** Following [36,45], we used the first frame image and the corresponding object masks as the user-provided query data. We also follow them and report region-based similarity $\mathcal{J}$ (the Jaccard Index, measuring the overlap between the predicted and ground truth regions), contour-based accuracy $\mathcal{F}$ (evaluating the accuracy of the predicted contour compared to the ground truth contour) and $\mathcal{J}\&\mathcal{F}$ as evaluation metrics.

Table 2: **Personalized Retrieval:** Mean Average Precision (mAP) on various benchmarks comparing PDM with state-of-the-art self-supervised, weakly supervised, and supervised methods. While our method yields superior performance, other methods leveraging our PDM features also yield a performance boost.

| Methods | ROxford Medium | ROxford Hard | RParis Medium | RParis Hard | PerMIR |
|---|---|---|---|---|---|
| **Self & Weakly Supervised** | | | | | |
| MAE [11] | 11.7 | 2.2 | 19.9 | 4.7 | - |
| iBOT [46] | 39.0 | 12.7 | 70.7 | 47.0 | - |
| DINOv2 [23] | 75.1 | 54.0 | 92.7 | 83.5 | 29.7 |
| CLIP [29] | 28.5 | 7.0 | 66.7 | 41.0 | 20.9 |
| OpenClip [13] | 50.7 | 19.7 | 79.2 | 60.2 | 26.7 |
| GLIP [18] | - | - | - | - | 31.2 |
| BLIP [17] | - | - | - | - | 33.3 |
| SLIP [22] | - | - | - | - | 35.9 |
| PDM (**ours**) | **77.2** | **58.3** | **93.4** | **84.7** | **73.0** |
| OpenClip + PDM (**ours**) | **70.1** | **57.7** | **90.1** | **82.0** | **69.9** |
| DINOv2 + PDM (**ours**) | **80.4** | **62.1** | **93.6** | **85.1** | **70.8** |
| **Supervised** | | | | | |
| GSS [21] | 80.6 | 64.7 | 93.4 | 85.3 | - |
| HP [2] | 85.7 | 70.3 | 92.6 | 83.3 | - |
| SuperGlobal [34] | 90.9 | 80.2 | 93.9 | 86.7 | 33.5 |
| GSS + PDM (**ours**) | 89.3 | 76.1 | 92.9 | 84.8 | **62.0** |
| SuperGlobal + PDM (**ours**) | **91.2** | **80.3** | **94.0** | **86.8** | **69.1** |

**Compared methods.** We compare our approach with various *self-supervised* foundation models: (1) DINOv2 [23], (2) PerSAM [45] and (3) DIFT [36] and DiffSeg [38]. We also compare with *SoTA supervised methods* that were trained on the task of video segmentation. Namely, MAST [15] and SFC [12].

**Results.** Table 1b presents the results of our experiments in the video label propagation task. Our method demonstrates competitive performance on the DAVIS [27] dataset and superior results on PerMIS benchmark. Our method achieves a notable balance between $J$ and $F$, indicating its effectiveness in capturing both region and contour details. Improvement in PerSAM(PDM) demonstrated that our PDM can boost results also for other methods.

## 5.3 Personalized Retrieval

**Datasets.** We conduct experiments across various retrieval benchmarks, including both single-instance and multi-instance datasets. Initially, we assess our model's performance on the widely-used $\mathcal{R}$Oxford and $\mathcal{R}$Paris datasets [25, 26] with revised annotations [28]. These datasets consist of 4,993 and 6,322 images, each featuring a single instance. Evaluation involves 70 query images per dataset, categorized into Easy, Medium, and Hard tasks based on retrieval complexity, with our focus primarily on the more challenging Medium and Hard tasks. Instance masks are obtained from [3]. We further evaluate our model on the PerMIR benchmark (Section 4).

**Baselines.** We compare our approach with state-of-the-art models, including self-supervised foundation models: MAE [11], SEER, and DINOv2 [23]; weakly-supervised foundation models: CLIP [29] and OpenClip [13]; and fully supervised methods: GSS [21], HP [2], and SuperGlobal [34]. Both self-supervised foundation models and weakly supervised foundation models were evaluated without further training or fine-tuning. We showcase results utilizing PDM both independently and as a re-ranking technique built upon various frozen pre-trained models (used for global feature retrieval). We denote this combination of methods, in Table 2 by the name of the pre-trained model + PDM. We follow [34] and apply re-ranking on the top 400 global features with the highest scores from the pre-trained model.

**Evaluation Protocol.** Following [23, 34], we report the mean average precision (mAP) for all methods. In all experiments, we used code and parameters provided by the authors of the compared methods.

**Results.** Table 2 presents the Mean Average Precision (mAP) across all benchmarks, highlighting the retrieval performance of PDM. Our method consistently outperforms all self-supervised and weakly supervised foundation methods and achieves comparable results to supervised methods. Notably, it surpasses DINOv2 [23] on the $\mathcal{R}$Oxford-hard dataset by $+4.3\%$ and by $+43\%$ on the PerMIR benchmark. Additionally, using PDM for reranking, we achieve better performance than SoTA-supervised methods, on the $\mathcal{R}$Paris and $\mathcal{R}$Oxford benchmarks. The results on the PerMIR benchmark underscore the inherent challenges faced by current methods in handling multi-instance samples. In contrast, our method demonstrates the robustness and effectively retrieves the correct samples, highlighting the efficacy of features derived from pre-trained diffusion models for instance-based retrieval tasks. Figure 5(b) provides qualitative retrieval results showing that our model successfully identifies images containing the same instance, while other methods primarily capture semantic similarity. See Appendix F for additional qualitative results.

## 6 Summary and Limitation

In this paper, we introduce a zero-shot approach for utilizing pre-trained Stable Diffusion (SD) features for personalized retrieval and segmentation tasks. We also review existing benchmarks for these tasks and propose a new benchmark to better evaluate performance. Our method showcases SoTA performance in three different personalization tasks. Nevertheless, it requires image inversion for feature extraction and therefore may depend on the success of image reconstruction quality.

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

# Appendix

## A Perliminaries

**Denoising module of text-to-image diffusion model.** We start by describing the different layers that compose the denoising module of a Text-to-Image diffusion model. Latent Diffusion Model [31], applies the diffusion process in the latent space of a pre-trained image autoencoder. This model adopts a U-Net [32] architecture conditioned on the guiding prompt $P$. The U-Net is composed of several layers where each consists of three types of blocks: (1) a residual block, (2) a self-attention block, and (3) a cross-attention block as illustrated in Figure 1. At each timestep of the denoising process, the noised latent code $z_t$ is fed as input to the U-net. **The residual block** convolves image features, $z_t$ to produce intermediate features $\phi(z_t)$. **In the self-attention block**, $\phi(z_t)$ projected into "queries"

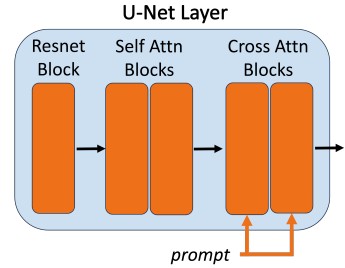

Fig. S 1: Single block of a U-Net layer ( Stable Diffusion [31]).

$Q$, "keys" $K$ and "values" $V$. For each query vector $q_{i,j}$, representing a patch, located at the spatial location $(i, j)$ of $\mathcal{Q}$, the self-attention map is then given by:

$$A_{(i,j)} = softmax(\frac{q_{i,j} \cdot \mathcal{K}^T}{\sqrt{d}}). \tag{8}$$

The last block, **the cross-attention block**, facilitates interaction between the spatial image features extracted from the self-attention block and the token embeddings of the text prompt $P$. The process is similar to that in the self-attention layer, but here, $\mathcal{Q}$ is derived from the spatial features of the previous self-attention layer, while $\mathcal{K}$ and $\mathcal{V}$ are projected from the token embeddings of the prompt.

## B Ablation Study

In this section, we ablate key components of our method.

**Personalized Retrieval.** *(1) Object Mask Instead of Class Name:* In this scenario, we considered the case where the class name is not provided, but an object mask is available. We tested this configuration on PerMIS, resulting in a mIOU of 45.0% compared to the original 42.3% when using the class name. The bIOU was 89.2% compared to the original 86.8% when using the class name. This shows that using an object mask leads to improved segmentation performance, indicating its potential as a valuable alternative when class names are not available. *(2) Appearance vs. Semantic Maps:* We examined the individual contributions of the Appearance and Semantic maps to the final similarity map. For this experiment, we used each map independently as the final similarity map, ignoring the other. When using only the Appearance Map, we achieved a mIOU of 30.2%, compared to 24.9% when using only the Semantic Map. Both results are significantly lower than our original mIOU of 42.3% when using both maps and averaging them. These findings underscore the necessity of integrating both maps to achieve optimal performance in the final similarity map, and eventually in personalized matching.

**Personalized Segmentation.** *(1) Object mask instead of class name*: Here we explore our approach when the input image is not accompanied by a class name but rather by a precise segmentation mask of the personalized object. During inversion, the prompt is set to be an empty string. The segmentation mask is used to distinguish the personalized object's features from the input image instead of cross attention map. We tested this configuration on PerMIR, resulting in a mAP of **76.2** compared to the original 73.0 when using the class name. This illustrates the strong capabilities of the semantic map obtained using the cross-attention layer. *(2) Appearance vs Semantic maps*: Here we examine the individual contributions of the Appearance and Semantic maps to the final similarity map $\mathcal{S}^{DF}$ calculated in our method. For this experiment, we use each map independently as the final similarity map $\mathcal{S}^{DF}$, ignoring the other (instead of averaging them, as explained in Section 3, Eq.(7)). When using only the Appearance Map, we achieve a mAP of **42.3**, compared to **32.9** when using only

Table S 1: Performance comparison across diffusion models. We report segmentation performance (mIoU, bIoU), feature extraction run time per image, and mean PSNR for the inversion-reconstruction quality.

| Diffusion Model | PerSeg | | PerMIS | | Feature Extraction | Mean |
| | mIoU | bIoU | mIoU | bIoU | Run Time (s) | PSNR |
| --- | --- | --- | --- | --- | --- | --- |
| SDXL-turbo | 95.4 | 79.8 | 42.3 | 86.8 | 0.5 | 24.1 |
| SDXL | 97.0 | 80.9 | 44.8 | 87.7 | 5 | 25.9 |
| SDv2.1 | 95.9 | 80.1 | 43.7 | 87.1 | 5 | 25.8 |

the Semantic Map. Both results are significantly lower than our original mAP of **73.0** when using both maps and averaging them. These findings underscore the necessity of integrating both maps to achieve optimal performance in the final similarity map $S^{DF}$.

## C   PerMIR and PerMIS Statistics

In this section, we describe the statistics of our newly introduced benchmark, Personalized Multi-Instance Retrieval (PerMIR). Following our image extraction process from the BURST dataset (detailed in Section 4), each video results in three different images of the personalized object, with each image containing an average of 3.1 different objects. We randomly select one image to serve as the query, while the other two are labeled as positive instances in the gallery. This process yields a total of **150** queries and a gallery comprising **450** images. The object distribution among the 150 query images is as follows: 51 persons, 52 cars, 10 animals, 4 food items, and 33 other objects (*e.g.* cup, drawer, tennis racket, slippers).

Random frame selection was done once during dataset preparation to ensure fair comparisons among all methods. We manually inspected the frames for quality and diversity, finding them acceptable and adequate given the BURST [4] dataset's quality and video length. We thus further quantified frames quality and diversity. Using the CLIP model, we found an average cosine similarity of 0.17 between frames, indicating low similarity (compared to 0.31 for adjacent frames) and thus high diversity. For quality, the mean SSIM between dataset frames and a random ImageNet subset was 13.2 (compared to 11.8 for ImageNet samples, higher values indicate better quality).

## D   Performance Across Diffusion Models

To evaluate how Personalized Diffusion Model (PDM) performance and quality vary with different diffusion models, we conducted experiments using three models: SDXL-turbo, SDXL, and SDv2.1. These experiments were performed on two personalized image segmentation datasets: PerSeg and PerMIS. For each diffusion model, we report segmentation performance in terms mIoU and bIoU, as well as the feature extraction run time per image and the mean PSNR of the inversion-reconstruction process.

Table 1 summarizes the results. It shows that while SDXL and SDv2.1 provide better performance in both mIoU and bIoU compared to SDXL-turbo, their inversion-reconstruction time is significantly longer, as these models require more inversion steps. Specifically, the reconstruction time for SDXL and SDv2.1 is 10 times slower than SDXL-turbo. Nevertheless, these models yield higher PSNR values, indicating better inversion-reconstruction quality.

As indicated by the results, PDM features can be found for other diffusion models like SDXL and SDv2.1, which yield better segmentation performance (higher mIoU and bIoU values) and improved reconstruction quality (higher PSNR) at the cost of longer inversion times. These findings further confirm the robustness of PDM features across different diffusion models.

This paper focused on UNet-based diffusion models because they are currently the most widely-used text-to-image models. We are optimistic that similar features can be found in other diffusion models, for the following reasons. First, recent studies [39] identified structural and appearance features in vision transformer-based models. Second, it was not hard to find instance-features in several Unet

Table S 2: Comparison of performance between feature averaging and weighted averaging for combining appearance and semantic features on the ROxford-Hard and PerMIR datasets.

| Method | ROxford-Hard | PerMIR |
|---|---|---|
| Feature Averaging (in the paper) | 53.2 | 71.2 |
| Weighted Averaging | 58.4 | 76.9 |

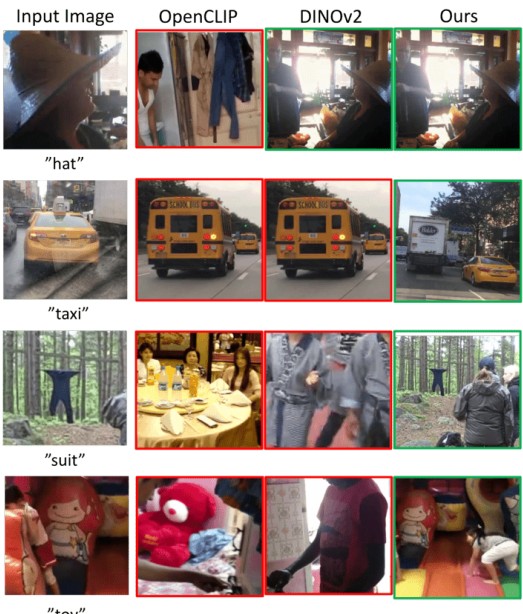

Fig. S 2: Qualitative examples for personalized retrieval: DINOv2 exhibits improved instance-based characteristics compared to OpenCLIP. However, unlike other methods that attend to the color or texture, our (PDM) method can leverage both semantic and appearance cues to successfully identify instances, even under substantial variations.

diffusion models as illustrated above. Thus, we assume that other diffusion models (such as DiTs) will also exhibit comparable or better instance features.

## E   Combining Appearance and Semantic Features

In the main paper, feature averaging was used to combine appearance and semantic features in order to avoid training or hyperparameter tuning on labeled data. We conducted a further analysis using a weighted combination of semantic and appearance features, optimized on a training set to explore more complex fusion methods.

For this, the PerMIR and ROxford-Hard datasets were split into 20% training and 80% test sets, and the weighted fusion parameters were optimized on the training sets. The results are summarized in Table 2. The weighted combination of features led to improvements in performance compared to simple averaging, with gains of 5.2% on the ROxford-Hard dataset and 5.7% on the PerMIR dataset.

These findings suggest that, when a training set is available, weighted fusion can significantly enhance performance. This opens up the potential for further exploration of more sophisticated, learnable fusion methods in future work.

These results highlight the benefits of weighted fusion for combining appearance and semantic features, and future work will investigate more advanced techniques that dynamically adapt to the data.

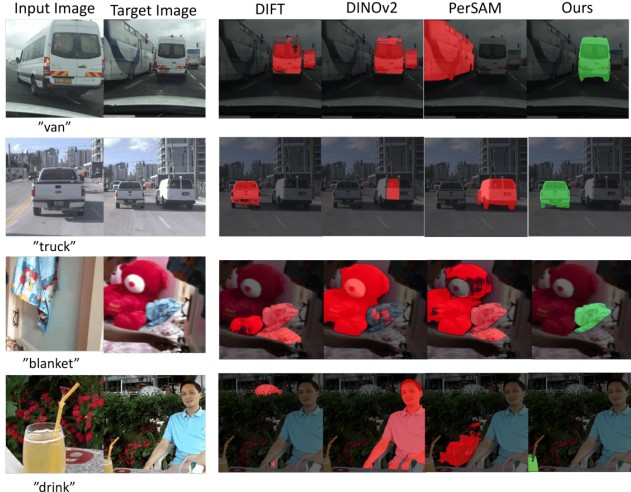

Fig. S 3: Qualitative examples for personalized segmentation: Rows 1,2 show cases where existing similar objects in the scene often distract previous features, while our proposed PDM successfully identifies and segments the correct instance. Note the successful segmentation of the small blanket (row 3) and substantially occluded drink (row 4).

## F    Additional Qualititive Results

We provide additional qualitative results for personalized retrieval and personalized segmentation. Figure S3 shows segmentation results on PerMIS and Figure S2 shows top-1 retrieved image of different methods on PerMIR.

