# OpenReview forum: "Where's Waldo: Diffusion Features For Personalized Segmentation and Retrieval"
_NeurIPS.cc/2024/Conference — NeurIPS 2024 poster_

### Official Review · Reviewer_Axpb · 2024-06-26

**Soundness:** 3
**Presentation:** 2
**Contribution:** 2
**Rating:** 4
**Confidence:** 4

**Summary:**

The authors propose a novel approach without any additional training for Personal retrieval and segmentation tasks, which focus on identifying specific instances within a dataset. The existing methods struggle to locate a desired instance when other instances within the same class are presented. They propose to utilize the features from a pre-trained text-to-image diffusion model and design a Personal Features Diffusion Matching method. The proposed method outperforms on both existing datasets and the new dataset which is proposed in this paper.

**Strengths:**

+ The idea of introducing the features from the text-to-image diffusion models to handle the personal retrieval and segmentation problem is interesting.
+ The authors propose a new approach called PDM which requires no training or fine-tuning, no prompt optimization, or any additional models.
+ The authors explore the hidden textural and semantic information in the layers and blocks in the diffusion model.
+ The authors construct a new benchmark includes videos with multiple instances from the same category.

**Weaknesses:**

- The logical flow before the Method Section of this paper needs to be adjusted. It is hard to understand what the authors want to do and the problem they want to address, although the utilization of features from diffustion models and the PDM itself is somewhat clear.
- The form of the task in the this paper is not clear.
	"... a dataset based on an input image and a short description of the reference instance." in Line.1-2
	"this work focuses on the case where only class names are provided." Line.102-103
	... Line.103-105
	As observed in the figures, the input information includes reference images, target images, and the class name (some of them are optional)
- I think the SUPERVISED models for personal retrieval and segmentation, which need a large amount of annotated data, and the SELF-/WEAKLY-SUPERVISED foundation models are not in the same aspect. The authors should rearrange the related works and the motivation of their work.
- The authors should list their contributions to help the readers better understand thier work.
- The visualized results in both the main text and appendix do not fully capture the problem the authors aim to address.  Few of them present the situation where a desired instance appears together with other instances of the same class.
- There is a lack of sufficient theoretical justification for why the authors choose diffusion models and why the features of diffusion models can achieve the capabilities that the authors expect. Only using "These models have the capability to generate an infinite array of objects and instances, each exhibiting unique appearances and structures. Line.33-34" and the two examples in Figure 2 is not enough.

**Questions:**

Please refer to the *Weakness* part.
Additional comments:
1. The authors need to clearly define the form of the task and formally present the settings for differernt scenarios.
2. The Figure 3 is hard to understand. Why "dog" is not fed into the diffusion features extraction in I_t branch? As F^s_t is features from the cross attention, what information is used to calculate the cross attention with I_t?
3. The authors should explain why using the features from diffusion models and provide theoretical justification. They should figure out the differences between the diffusion models and the other models (e.g. common resnet, ViT, Vision Mamba,  Transformer+KAN, ..., and the other variants, why not choose other SoTA foundation models). Why they can present the textural and semantic information?

---

> ### Author Rebuttal · Authors · 2024-08-05
>
> Thank you for finding our approach novel and the ides of our method interesting. We address your comments below.
>
> **Q1: The logical flow before the method section needs to be adjusted. The authors should rearrange the related works and the motivation of their work. The authors should list their contributions.**
> **A1:** We appreciate the reviewer's valuable feedback and their contribution to improving our paper. We will make sure to rearrange the related works and the motivation of our work to improve the logical flow leading up to the method section. We will make a clear distinction between supervised and self/weakly supervised foundation models in the text and tables. We will also explicitly list our contributions to highlight the key aspects and innovations of our work.
>
> **Q2: It is hard to understand what the authors want to do and the problem they want to address.**
> **A2:** This paper addresses Personalized Retrieval and Personalized Segmentation, where a user provides a reference image and either a mask or the object's class name  (lines 1-2, 101-102 in the paper). Personalized retrieval aims to find database images with the exact object from the reference image, while personalized segmentation seeks to segment the specified object in new images and videos. We demonstrate that features from pre-trained diffusion models achieve high zero-shot performance on these tasks without training or fine-tuning (please refer to Section 3.1 for details). Lastly, we further point to major drawbacks with current personalized retrieval and segmentation benchmarks and offer two new alternative benchmarks. We show that our approach (PDM) suppresses all supervised and self-supervised methods on all benchmarks. We will make sure that these issues are better emphasized in the revised version.
>
> **Q3: What is the “short description” mentioned in the paper?**
> **A3:** The “short description” is  the “class name” (as we use in lines 1-2, 101-102, 124, 157). We will make sure to clarify this in the final version.
>
> **Q4: Visualized results do not present the situation where a desired instance appears together with other instances of the same class.**
> **A4:** Due to space constraints, we cropped the images in the figures to focus on the relevant objects among other similar objects. We will include more examples without cropping and enlarge the images in the final version.
>
> **Q5: Why using diffusion features and not other models?**
> **A5:** The fundamental difference between diffusion models and the other models listed by the reviewer is that they are generative, in the sense that they focus on p(image|text). The diffusion training objective (i.e., coarse-to-fine reconstruction loss) requires the model to *generate* informative features for every object. This is different from other, non-generative models, some of which use image-level contrastive learning objectives (e.g. CLIP), and others use image-level discriminative objectives (Resnet, ViT).  Such image-level discriminative or contrastive objectives often remove information about specific instances and objects [4]. Furthermore, current diffusion models contain a self-attention component which controls how some parts of the image depend on other parts. This allows the model to generate coherent objects and makes the model contain an explicit representation of object instances.  For additional details, please refer to recent papers on diffusion features [1,2,3]. We will clarify and add this discussion to the final version.
>
> **Q6: Figure 3 is hard to understand. Why "dog" is not fed into the target branch?**
> **A6:** We thank the reviewer for their feedback. We will make sure to simplify the figure in the final version. This figure illustrates the flow of our algorithm and aligns with the algorithm described in Section 3. Colors are used to differentiate between "Appearance" (green) and "Semantic" (yellow) features. In this figure, a reference image of a dog labeled "dog" and a target image are provided. Both images and the class name undergo feature extraction (see L155-156).  The reference and target branches follow the same process, making it irrelevant which branch receives the class name.
>
> **Q7: What information is used to calculate the cross attention with the target image?**
> **A7:** Semantic similarity is calculated as the cross attention between class name token $C$ and the semantic feature map $F^{S}_{t}$ as detailed in lines 170-172 and Eq. 6.
>
>
> [1] Tang et al. (2023) “Emergent Correspondence from Image Diffusion”.
> [2] Lue et al. (2023) “Diffusion Hyperfeatures: Searching Through Time and Space for Semantic Correspondence”.
> [3] Zhang et al. (2023) “A Tale of Two Features: Stable Diffusion Complements DINO for Zero-Shot Semantic Correspondence”.

---

> > ### Comment · Reviewer_Axpb · 2024-08-13
> >
> > Thank you for the detailed explanation. After reading the authors' response, most of my concerns have been resolved. The motivation and proposed method in this work are interesting. The presentation could be further improved to some extent. However,  after reading the comments from other reviewers, I have the same problems as Reviewer 8kmZ. PDM has a weakness in terms of inference speed and is limited by the robustness of the used diffusion model, especially as generative models struggle to generate certain types of objects effectively.

---

> ### Author Response · Authors · 2024-08-13
>
> Thank you for your feedback.
> Regarding the robustness of diffusion features, our experiments show they are effective across a wide range of concepts, outperforming ***both supervised and self-supervised methods on various benchmarks***. What tools would the reviewer suggest to further validate the robustness?
> We believe that as better and faster diffusion models are released, PDM can be applied to achieve more robust features and thus better performance. Introducing PDM will indeed allow further research in this direction.
> As for the inference speed, it is indeed a limitation as noted in our paper. We currently achieve feature extraction in half a second, making it applicable to a variety of tasks. As inversion methods improve, our approach will also speed up.

---

### Official Review · Reviewer_CWZR · 2024-07-01

**Soundness:** 4
**Presentation:** 4
**Contribution:** 3
**Rating:** 7
**Confidence:** 4

**Summary:**

The paper introduces Personalized Diffusion Features Matching (PDM), a novel zero-shot approach that utilizes pre-trained text-to-image diffusion models for personalized image retrieval and segmentation without requiring additional training. PDM extracts and fuses semantic and appearance features from an off-the-shelled diffusion model to accurately identify and segment unique instances, even when multiple similar objects are present. It demonstrates superior performance on various benchmarks, outperforming both self-supervised and supervised methods. The authors also address limitations in current datasets by proposing new benchmarks that include challenging scenarios with multiple instances from the same category. Despite its reliance on image inversion, which may affect performance, PDM offers a significant advancement in the field of personalized instance retrieval and segmentation.

**Strengths:**

1. The paper is well organized, clearly written, and easy to follow, especially for the motivation clarification.
2. The experimental observation and justification of the effect of self-attention and cross-attention in the diffusion model are insightful.
3. The proposed method is technically sound and well-motivated.
4. The contribution is sound, consisting of new observations, methods, and more practical benchmark settings.
5. The proposed method is training-free and seems to have great transferability.

**Weaknesses:**

I didn't find a major weakness

**Questions:**

1. It is suggested that some works related to semantic-level feature matching and their applications [1,2] be added to the related work section.

2. (Just constructive suggestions for future works). I am curious whether a similar or more profound phenomenon about attention will appear in DiT [3] and stable video diffusion [4]. Can the method be extended to these more advanced architectures [3,4]?


[1] Lindenberger, P., Sarlin, P. E., & Pollefeys, M. (2023). Lightglue: Local feature matching at light speed. In Proceedings of the IEEE/CVF International Conference on Computer Vision (pp. 17627-17638).

[2] Li, W., Liu, X., & Yuan, Y. (2022). Sigma: Semantic-complete graph matching for domain adaptive object detection. In Proceedings of the IEEE/CVF Conference on Computer Vision and Pattern Recognition (pp. 5291-5300).

[3] Peebles, W., & Xie, S. (2023). Scalable diffusion models with transformers. In Proceedings of the IEEE/CVF International Conference on Computer Vision (pp. 4195-4205).

[4] Blattmann, A., Dockhorn, T., Kulal, S., Mendelevitch, D., Kilian, M., Lorenz, D., ... & Rombach, R. (2023). Stable video diffusion: Scaling latent video diffusion models to large datasets. arXiv preprint arXiv:2311.15127.

---

> ### Author Rebuttal · Authors · 2024-08-05
>
> Thank you for finding our approach insightful, technically sound with great transferability and our paper to be well motivated and easy to follow. We kindly address your comments below.
>
> **Q1: Cite relevant work.**
> **A1:** Thank you for your feedback. We will make sure to cite these papers in the final version.
>
> **Q2: Can the method be extended to DiT and stable video diffusion?**
> **A2:** We thank the reviewer for their suggestion. We are optimistic that PDM can be applied to DiT and video diffusion models. First, recent studies [1] identified structural and appearance features in vision transformer-based models. Second, it was not hard to find instance-features in several Unet models, SDXL and SDv2.1. Thus, we assume that other diffusion models will also exhibit comparable instance features. We appreciate the reviewer's insightful question, and we leave this for future work.
>
> [1] Tumanyan et al. (2022) “Splicing ViT Features for Semantic Appearance Transfer”.

---

> > ### Comment · Reviewer_CWZR · 2024-08-12
> >
> > Thanks for the clarification. I have read all the responses. The proposed method can perform better after fine-tuning, which shows more significant potential in benchmark performance. Considering the new observations, clear motivations, methodology designs, and dataset contributions, I would like to keep my original score because this can be seen as a great contribution to the prior work DIFT that explores diffusion features.

---

> > > ### Author Response · Authors · 2024-08-14
> > >
> > > Dear reviewer, Thank you for the positive and insightful feedbacks

---

### Official Review · Reviewer_JGDB · 2024-07-03

**Soundness:** 3
**Presentation:** 3
**Contribution:** 3
**Rating:** 5
**Confidence:** 5

**Summary:**

This paper explores the use of text-to-image diffusion models for personalized retrieval and segmentation tasks. The authors introduce a novel method called PDM (Personalized Features Diffusion Matching), which leverages intermediate features from pre-trained text-to-image models for personalization tasks without requiring additional training. PDM demonstrates superior performance across multiple benchmarks, surpassing many supervised methods. Additionally, the paper identifies deficiencies in existing datasets and introduces a new benchmark dataset that effectively addresses the challenges presented by traditional datasets in scenarios involving multiple instances of the same class. This new dataset focuses on complex scenes containing multiple instances of the same class, further validating PDM's efficiency and accuracy in precisely handling and locating multiple similar instances.

**Strengths:**

1.This paper introduces a novel challenge in personalized retrieval and segmentation tasks involving complex scenes with multiple instances of the same class. In response to this scenario, the authors have constructed a new benchmark dataset and devised a novel method—PDM (Personalized Features Diffusion Matching), which ingeniously combines appearance and semantic features to address the issue of multiple instances within the same category. This approach is quite innovative.

2.The overall structure of the paper is logically organized and easy to understand, and the figures greatly assist in comprehension. Overall, the paper is excellently written.

**Weaknesses:**

1.The authors simply take the average of appearance and semantic features as the overall diffusion features without considering a weighted fusion of these two features. This approach may lead to errors in segmentation or retrieval on some datasets, as it fails to balance the contribution between different features effectively.

2.There may be some issues with the process of generating the personalized multi-instance retrieval and segmentation dataset, such as the randomness in selecting video frames. While this approach improves the efficiency of data generation, it could mislead model training and evaluation if the chosen frames lack sufficient feature diversity or have quality issues.

3.Although the paper conducts extensive experiments in personalized retrieval and segmentation tasks and briefly supplements with ablation study results of key components in the appendix, and qualitatively analyzes the differences from other methods, ensuring comprehensive experimentation, there are still shortcomings. The ablation experiments are only conducted for multi-instance retrieval, which does not prove the importance of key components in multi-instance segmentation. It would be beneficial to include experiments addressing this aspect as well.

**Questions:**

1.Given your mention that the method's performance might depend on the success of image reconstruction quality, I am curious about how you have managed to enhance image reconstruction quality to achieve such impressive results with your approach.
2.Have you considered more complex ways of combining appearance and semantic features? If so, why did these approaches not yield better results, and what are your thoughts on this?
3.Why opt for random selection of video frames without addressing potential quality issues in the generated data? How do you manage this randomness to ensure fairness and consistency across different methods tested on this data?

**Limitations:**

The authors briefly discuss the limitations of their method in the conclusion but do not provide any viable future solutions. Could the authors possibly offer some feasible ideas or directions for addressing these limitations in future work?

---

> ### Author Rebuttal · Authors · 2024-08-05
>
> Thank you for your feedback. We are encouraged that you find our approach innovative, the challenge novel and our paper to be excellently written. We address your comments below.
>
> **Q1: Have you considered more complex ways of combining appearance and semantic features?**
> **A1:** We thank the reviewer for their suggestion.  In the paper, we chose feature averaging to avoid training or hyperparameter tuning on labeled data. Following the reviewer's suggestion, we tested a weighted combination of semantic and appearance features: $wF_{appearance} + (1-w)F_{semantic}$.  We split the PerMIR and ROxford-Hard [1] datasets into training (20%) and test sets (80%), and optimized $w$ on the training sets. The table below shows that when a training set is available, weighted fusion improves results. We also plan to explore learnable fusion methods with advanced techniques in future work. We will include this analysis in the final version.
>
>
> |               | ROxford-Hard | PerMIR |
> |---------------|--------------|--------|
> | Avg (in the paper)           | 53.2         | 71.2   |
> | Wegithed Avg  | **58.4**         | **76.9**   |
>
>
> **Q2: In the proposed multi-instance benchmarks, selected frames may lack feature diversity or suffer from quality issues. How do you ensure fairness and consistency across methods despite randomness?**
> **A2:** We appreciate the reviewer’s comment. Random frame selection was done once during dataset preparation to ensure fair comparisons among all methods. We manually inspected the frames for quality and diversity, finding them acceptable and adequate given the BURST [2] dataset’s quality and video length. In response to this comment, we further quantified frames quality and diversity. Using the CLIP model, we found an average cosine similarity of 0.17 between frames, indicating low similarity (compared to 0.31 for adjacent frames) and thus high diversity. For quality, the mean SSIM between dataset frames and a random ImageNet subset was 13.2  (compared to 11.8  for ImageNet samples, higher values indicate better quality). We will clarify and include this analysis in the final version.
>
> **Q3: Missing ablation study for Personalized Segmentation.**
> **A3:** Following this comment, we now added the following ablation study for personalized segmentation:
> *(1) Object Mask Instead of Class Name:* In this scenario, we considered the case where the class name is not provided, but an object mask is available. We tested this configuration on PerMIS, resulting in a mIOU of 45.0% compared to the original 42.3% when using the class name. The bIOU was 89.2% compared to the original 86.8% when using the class name. This shows that using an object mask leads to improved segmentation performance, indicating its potential as a valuable alternative when class names are not available.
>
> *(2) Appearance vs. Semantic Maps:* We examined the individual contributions of the Appearance and Semantic maps to the final similarity map. For this experiment, we used each map independently as the final similarity map, ignoring the other. When using only the Appearance Map, we achieved a mIOU of 30.2%, compared to 24.9% when using only the Semantic Map. Both results are significantly lower than our original mIOU of 42.3% when using both maps and averaging them. These findings underscore the necessity of integrating both maps to achieve optimal performance in the final similarity map, and eventually in personalized matching.
> We thank the reviewer for this feedback. We will make sure to add this ablation study to the final version.
>
> **Q4: How did you manage to enhance image reconstruction quality?**
> **A4:** We simply build on recent advances in inversion with diffusion models [3, 4]. For SDXL-turbo, using [3] for inversion results in a PSNR of 24.1, close to the upper bound of 26.3 set by the diffusion VAE. This high PSNR indicates good reconstruction quality and hence high-quality features.
>
> **Q5: Could the authors suggest ways to address the limitations in future work?**
> **A5:** Image inversion is the primary limitation of our method, and thus it falls into a different line of work. We did not focus on developing new inversion techniques within the scope of this study. Note that our method is agnostic to the inversion approach, allowing us to adopt faster and more accurate methods developed by the community in the future.
>
> [1] Radenovic et al. (2018) “Revisiting oxford and paris: Large-scale image retrieval benchmarking”.
> [2] Athar et al. (2023) “Burst: A benchmark for unifying object recognition, segmentation and tracking in video”.
> [3] Pan et al. (2023) “Effective real image editing with accelerated iterative diffusion inversion”.
> [4] Garibi et al. (2024) “ReNoise: Real Image Inversion Through Iterative Noising”.

---

> > ### Comment · Reviewer_JGDB · 2024-08-12
> > **Official Comment by Reviewer JGDB**
> >
> > Thanks for your detailed response. I would keep my score by a comprehensive consideration of the paper's contribution and  experimental results.

---

> > > ### Author Response · Authors · 2024-08-14
> > >
> > > Dear reviewer, Thank you for the positive and insightful feedbacks

---

### Official Review · Reviewer_8kmZ · 2024-07-05

**Soundness:** 3
**Presentation:** 3
**Contribution:** 2
**Rating:** 4
**Confidence:** 2

**Summary:**

This paper introduces a novel approach called Personalized Diffusion Features Matching (PDM) for personalized retrieval and segmentation tasks. Most current self-supervised and supervised methods struggle to accurately identify specific instances within a dataset when multiple instances from the same class are present. So the proposed PDM method leverages intermediate features from pre-trained text-to-image diffusion models. PDM combines semantic and appearance cues to accurately locate and segment specific instances within target images.

**Strengths:**

* its an innovate use of pretrained model (already available foundation models ) in a zero shot setting. This is beneficial for people without a lot of compute and good for environment.

* the proposed method combines both appearance and semantic similarity, which is a well motivated design

**Weaknesses:**

* the authors used features from stable diffusion, which is text guided image generation model. its not that fair to compare to DINOv2 or SAM since those models were not trained with text supervision.

* the method depends on diffusion inversion, which is quite slower than other segmentation methods. the authors used sdxl-turbo to mitigate the latency but its unclear the impact on quality

* the method seems only applicable to Unet diffusion models. its unclear if it works on DiTs

**Questions:**

* what is the performance difference if PDM uses SDXL-turbo vs non-turbo? SDXL-turbo was notorious for not able to reproduce good quality details such as small human faces or limbs.

* how does PDM performance vary wrt to the strength of the underlying diffusion model?

* in table 2, the authors show openCLIP+PDM and DINOv2+PDM. what happens if you compute OpenCLIP+DINOv2+PDM?

**Limitations:**

yes the authors adequately address the limitations.

---

> ### Author Rebuttal · Authors · 2024-08-05
>
> Thank you for your detailed and insightful feedback. We believe we address all your concerns below.
>
> **Q1: It is not fair to compare PDM to Dinov2 or SAM since those models were not trained with text supervision.**
> **A1:** We thank the reviewer for the comment. The paper compared two types of baselines: (1) Baselines trained ***with*** text supervision (CLIP and OpenCLIP; See Table 2, Figure 5b and Figure S2) and (2) baselines without supervision, to be consistent with the literature [1,2,3].
> Following this comment, we add comparisons with more methods trained with text supervision. Specifically, we evaluated personalized retrieval with features from  BLIP2 [4], GLIP [5], and SLIP [6], on the PerMIR dataset. The table below shows that PDM outperforms all text-image models by a large margin. We will add this experiment and additional details in our revised manuscript/suppl.
> | Model                  | mAP  |
> |------------------------|------|
> | CLIP (in the paper)         | 20.9 |
> | OpenCLIP (in the paper)| 26.7 |
> | GLIP [5]                         | 31.2 |
> | BLIP [4]                         | 33.3 |
> | SLIP [6]                         | 35.9 |
> | **PDM (ours)**             | **73.0** |
>
>
> **Q2: How does PDM performance and quality vary with the underlying diffusion model?**
> **A2:** Thank you for the insightful question. Indeed, it is not clear a-priori that similar features with comparable quality can be found in other diffusion models. To answer this question, we repeated the feature finding process in two diffusion models: SDXL, and SDv2.1.
> We then ran new experiments for the Personalized Image Segmentation task using two datasets, PerSeg [7] and PerMIS. The table below reports segmentation performance using three diffusion models: (1) SDXL-turbo (results taken from the paper), (2) SDXL, and (3) SDv2.1. We also report the run time for feature extraction per image and the mean PSNR of images to measure inversion-reconstruction quality.
>
> The results show that one can find PDM features for other diffusion models (SDv2.1 and SDXL) that produce better results than SDXL-turbo in terms of PSNR, mIoU and bIoU. Their inversion reconstruction time is 10x slower because they require more inversion steps. We will report and discuss these results in the final version.
>
>
> | | PerSeg | | PerMIS | | | |
> |---|---|---|---|---|---|---|
> | | mIoU | bIoU | mIoU | bIoU | feature extraction run time / image (sec) | mean PSNR |
> | SDXL-turbo (in the paper) | 95.4 | 79.8 | 42.3 | 86.8 | 0.5| 24.1 |
> | SDXL | 97.0 | 80.9 | 44.8 | 87.7 | 5| 25.9 |
> | SDV2.1 | 95.9 | 80.1 | 43.7 | 87.1 | 5| 25.8 |
>
>
>
> **Q3: The method seems only applicable to Unet diffusion models.**
> **A3:** The paper focused on UNet-based diffusion models because they are currently the most widely-used text-to-image models.  We are optimistic that similar features can be found in other diffusion models, for the following reasons. First, recent studies [8] identified structural and appearance features in vision transformer-based models. Second, it was not hard to find instance-features in several Unet diffusion models. Thus, we assume that other diffusion models (such as DiTs) will also exhibit comparable or better instance features.
> We appreciate the reviewer's insightful question, as it highlights a promising avenue for future research.
>
>
> **Q4: What happens if you compute OpenCLIP+DINOv2+PDM?**
> **A4:** Following this question, we tested a combination of DINOv2 and PDM. Naturally, there are many ways to combine several features, and here we followed the protocol from [9,10]. We first used OpenCLIP to select the 400 most similar images to each query image. Then, we selected 100 images with highest DINOv2 scores, and then ranked them using PDM. We tested the combined method for personalized retrieval in two datasets: PerMIR and ROxford-Hard.
> In PerMIR, this method achieved a mAP of 70.2%, compared to 69.9% for OpenCLIP + PDM and 70.8% for Dinov2 + PDM. In ROxford-Hard, it achieved a mAP of 58.3%, compared to 57.7% for OpenCLIP + PDM and 62.1% for Dinov2 + PDM. This approach shows a slight improvement over OpenCLIP + PDM but underperforms compared to Dinov2 + PDM, likely due to OpenCLIP's less effective retrieval capabilities.
>
>
> [1] Tang et al. (2023) “Emergent Correspondence from Image Diffusion”.
> [2] Lue et al. (2023) “Diffusion Hyperfeatures: Searching Through Time and Space for Semantic Correspondence”.
> [3] Zhang et al. (2023) “A Tale of Two Features: Stable Diffusion Complements DINO for Zero-Shot Semantic Correspondence”.
> [4] Li et al. (2023) “BLIP-2: Bootstrapping Language-Image Pre-training with Frozen Image Encoders and Large Language Models”.
> [5] Li et al. (2022) “Grounded Language-Image Pre-training”.
> [6] Mu et al. (2022) “SLIP: Self-supervision meets Language-Image Pre-training”.
> [7] Zhang et al. (2023) “Personalize Segment Anything Model with One Shot”.
> [8] Tumanyan et al. (2022) “Splicing ViT Features for Semantic Appearance Transfer”.
> [9] Shao et al. (2023) “Global Features are All You Need for Image Retrieval and Reranking”
> [10] Zhu et al. (2023) “R2Former: Unified Retrieval and Reranking Transformer for Place Recognition”

---

> > ### Comment · Reviewer_8kmZ · 2024-08-10
> >
> > thank you for the detailed rebuttal

---

> > > ### Author Response · Authors · 2024-08-12
> > >
> > > Thank you for reading our response and for your feedback. We believe that it should have addressed your main concerns. If there are any remaining (or new) issues, we would love to get a chance to address them.

---

### Author Rebuttal · Authors · 2024-08-05

Dear Reviewers and ACs,
We were happy to see that reviewers found our approach **“novel”**, **“innovative”** (**All**), **“well-motivated"** (**R1, R3**) and recognized its potential as a **“significant advancement in the field of personalized instance retrieval and segmentation”** (**R3**). Additionally, they acknowledged our tasks and benchmarks as **“novel”** (**R1,R3**), **“practical”**, and **“challenging”** (**R1, R2, R3**), and found our method to be **“insightful”**, **“easy-to-follow”** and **“excellently written”**, with **“superior performance”** (**R2, R3**) compared to current SoTA approaches.
We have addressed the reviewers' concerns in our rebuttal and are open to further discussion.
Your input has been instrumental in improving our paper.

---

### Decision · Program_Chairs · 2024-09-25

**Decision:**

Accept (poster)

**Comment:**

The reviewers agree on the contribution, but had some differences on whether the limitations in computation time and potential coverage would limit the approach.  This was discussed in the rebuttal process and did not seem problematic enough to reject the paper.  The authors are encouraged to clarify the extent of these limitations and other reviewer comments.